# Memory Effects in High-Dimensional Systems Faithfully Identified by Hilbert–Schmidt Speed-Based Witness

**DOI:** 10.3390/e24030395

**Published:** 2022-03-12

**Authors:** Kobra Mahdavipour, Mahshid Khazaei Shadfar, Hossein Rangani Jahromi, Roberto Morandotti, Rosario Lo Franco

**Affiliations:** 1Dipartimento di Ingegneria, Università di Palermo, Viale delle Scienze, 90128 Palermo, Italy; kobra.mahdavipour@unipa.it (K.M.); mahshid.khazaeishadfar@unipa.it (M.K.S.); 2INRS-EMT, 1650 Boulevard Lionel-Boulet, Varennes, QC J3X 1S2, Canada; roberto.morandotti@inrs.ca; 3Physics Department, Faculty of Sciences, Jahrom University, Jahrom P.O. Box 74135111, Iran

**Keywords:** non-Markovianity, Hilbert–Schmidt speed, high-dimensional system, multipartite open quantum systems, memory effects

## Abstract

A witness of non-Markovianity based on the Hilbert–Schmidt speed (HSS), a special type of quantum statistical speed, has been recently introduced for low-dimensional quantum systems. Such a non-Markovianity witness is particularly useful, being easily computable since no diagonalization of the system density matrix is required. We investigate the sensitivity of this HSS-based witness to detect non-Markovianity in various high-dimensional and multipartite open quantum systems with finite Hilbert spaces. We find that the time behaviors of the HSS-based witness are always in agreement with those of quantum negativity or quantum correlation measure. These results show that the HSS-based witness is a faithful identifier of the memory effects appearing in the quantum evolution of a high-dimensional system with a finite Hilbert space.

## 1. Introduction

The unavoidable interaction of quantum systems with their environments induces decoherence and dissipation of energy. Recently, because of important developments in both theoretical and experimental branches of quantum information theory, studies of memory effects (non-Markovianity) during the evolution of quantum systems have attracted much attention (see Refs. [1,2,3] for some reviews). Some approaches used for a quantitative description of non-Markovian processes are either related to the presence of information backflows [4] or to the indivisibility of the dynamical map [5]. However, while well-defined for classical evolution, the notion of non-Markovianity appears to still lack a unique definition in the quantum scenario [6].

Non-Markovian processes, exhibiting quantum memory effects, have been characterized and observed in various realistic systems such as quantum optical systems [7,8,9,10,11,12], superconducting qubits [13,14], photonic crystals [15,16,17], light-harvesting complexes [18], and chemical compounds [19,20]. Moreover, it is known that non-Markovianity can be a resource for quantum information tasks [21,22,23,24,25]. Accordingly, various witnesses have been proposed to identify non-Markovianity based on, for example, distinguishability between evolved quantum states of the system [4], fidelity [26,27,28], quantum relative entropies [29,30], quantum Fisher information [31], capacity measure [32,33,34] and Bloch volume measure [35,36,37].

It has been shown that the nonmonotonic behavior of quantum resources such as entanglement [5], quantum coherence [38,39,40,41] and quantum mutual information [42] can be interpreted as a witness of quantum non-Markovianity. Using entanglement to witness non-Markovianity was first proposed in Ref. [5]. This proposal has been theoretically investigated for qubits coupled to bosonic environments [43,44,45], for a damped harmonic oscillator [46], and for random unitary dynamics and classical noise models [47,48,49]. It is also shown that entanglement cannot capture all the quantumness of correlations because there are some separable mixed states with vanishing entanglement, which can nevertheless have have nonzero quantum correlations [50]. Therefore, quantum correlations are more robust than entanglement [51,52,53,54], while entanglement may suffer sudden death [55,56]. Consequently, many methods to quantify quantum correlations have been provided, among which quantum discord [57,58] and measurement-induced disturbance [59] are proper for any bipartite state.

Recently, Hilbert–Schmidt speed (HSS) [60], a measure of quantum statistical speed which has the advantage of avoiding diagonalization of the evolved density matrix, has been proposed and employed as a faithful witness of non-Markovianity in Hermitian systems [61,62,63,64] and an efficient tool in quantum metrology [65,66]. These studies are so far especially limited to low-dimensional systems, while high-dimensional ones have not been investigated in detail. We know that high-dimensional systems play a crucial role in increasing the security in quantum cryptography [67,68], as well as in enhancing quantum logic gates, fault-tolerant quantum computation and quantum error correction [69]. This motivates us to check the sensitivity of HSS-based witness to detect non-Markovianity in high-dimensional and multipartite open quantum systems.

In this work, we analyze the validity of our HSS-based witness in various examples of high-dimensional open quantum systems with finite Hilbert spaces, such as qudits and hybrid qubit–qutrit systems. In particular, we consider a single qudit (spin-S systems) subject to a squeezed vacuum reservoir [70], and hybrid qubit–qutrit system coupled to quantum as well as classical noises [71]. We observe that the HSS-based witness is consistent with established non-Markovianity quantifiers based on dynamical breakdown of monotonicity for the quantum information resources.

The paper is organized as follows: In Section 2, we briefly review the definition of quantifiers. In Section 3, the sensitivity of HSS-based witness in high-dimensional and multipartite open quantum systems with finite Hilbert spaces through various examples is studied. Finally, Section 4 summarizes the main results and prospects.

## 2. Preliminaries

In this section, we briefly review the relevant quantifiers and concepts employed in this paper.

### 2.1. Non-Markovinity Definition

A classical *Markov process* is described by a family of random variables {X(t),t∈I⊂R}, for which the probability that *X* takes a value $x_{n}$ at any arbitrary time $t_{n}\in I$, provided that it took value $x_{n-1}$ at some previous time $t_{n-1}<t_{n}$, can be determined uniquely and may not be influenced by the possible values of *X* at times prior to $t_{n-1}$. It can be formulated in terms of conditional probabilities as follows: $P\left(x_{n},t_{n}|x_{n-1},t_{n-1};\ldots ;x_{0},t_{0}\right)=P\left(x_{n},t_{n}|x_{n-1},t_{n-1}\right)$ for all $\{t_{n}\geq t_{n-1}\geq ...\geq t_{0}\}\subset I$. Roughly speaking, its concept is connected with the memorylessness of the process and informally encapsulated by the statement that “a Markov process has no memory of the history of past values of *X*, i.e., the future of the process is independent of its history”.

To achieve a similar formulation in the quantum scenario we should find a way to define $P(x_{n},t_{n}|x_{n-1},t_{n-1};\ldots ;x_{0},t_{0})$ for quantum systems. In the classical realm, we may sample a stochastic variable without affecting its posterior statistics. However, ’sampling’ a quantum system requires measuring process, and hence disturbs the state of the system, affecting the subsequent outcomes. Therefore, $P(x_{n},t_{n}|x_{n-1},t_{n-1};\ldots ;x_{0},t_{0})$ depends on not only the dynamics but also the measurement process. Since in such a case the Markovian character of a quantum dynamical system is dependent on the the measurement scheme, chosen to obtain $P(x_{n},t_{n}|x_{n-1},t_{n-1};\ldots ;x_{0},t_{0})$, a definition of quantum Markovianity in terms of which is a challenging task. In fact, a reliable definition of quantum Markovianity should be independent of what is required to verify it.

The aforesaid problem may be solved by adopting a different approach focusing on studying one-time probabilities P(x,t). For these, in *linear* quantum evolutions, the definition of Markovianity reduces to the concept of *divisibility* defined without any explicit reference to measurement processes in the quantum scenario [1]. To introduce the divisibility concept, let us assume that the inverse of a quantum dynamical map $E_{t}$ exists for all times t≥0. Then it is possible to define a two-parameter family of maps by means of $E_{t,s}=E_{t}E_{s}^{-1}\;\;\left(t\geq s\geq 0\right)$ such that $E_{t,0}=E_{t}$ and $E_{t,0}=E_{t,s}E_{s,0}$. It should be noted that the existence of the inverse for all positive times guarantees the possibility of introducing the notion of divisibility, while $E_{t,0}$ and $E_{s,0}$ are required to be completely positive by construction, the map $E_{t,s}$ need not be completely positive and not even positive. It stems from the fact that the inverse $E_{s}^{-1}$ of a completely positive map $E_{s}$ need not be positive. The family of dynamical maps is called (C)P divisible when $E_{t,s}$ is (completely) positive for all t≥s≥0.

The trace norm given by $\left\|\rho \right\|=\mathrm{Tr}\sqrt{\rho^{†}\rho }=\sum_{k}\sqrt{a_{k}}$, in which $a_{k}$’s represent the eigenvalues of $\rho^{†}\rho$, leads to an important measure called *trace distance*, $D\left(\rho^{1},\rho^{2}\right)=\frac{1}{2}\left\|\rho^{1}-\rho^{2}\right\|$, for the distance between two quantum states $\rho^{1}$ and $\rho^{2}$. The trace distance $D(\rho^{1},\rho^{2})$ is interpreted as the *distinguishability* between states $\rho^{1}$ and $\rho^{2}$. Moreover, it is *contractive* for any completely positive and trace preserving (CPTP) map E affecting two arbitrary quantum states $\rho^{1,2}$, i.e., $D\left(E\left(\rho^{1}\right),E\left(\rho^{2}\right)\right)\leq D\left(\rho^{1},\rho^{2}\right)$ [3]. Because the dynamics of an open quantum system is described by a CPTP map $E_{t}$, the trace distance between the initial states is always larger than the trace distance between the time-evolved quantum states. Nevertheless, this fact does *not* mean that $D\left(\rho^{1}\left(t\right),\rho^{2}\left(t\right)\right)$, in which $\rho^{1,2}\left(t\right)\equiv E_{t}\left(\rho^{1,2}\left(0\right)\right)$, exhibits a monotonically decreasing function versus time [72].

There are various ways to define and detect non-Markovianity or memory effects in quantum mechanics (see [1] for a review). In Refs. [4,29], Breuer–Laine–Piilo (BLP) proposed one of the most well-known approaches, based on the variation in distinguishability of quantum states, to characterize the non-Markovian feature of the system dynamics. This is the non-Markovianity definition which we mention in our paper. According to BLP measure, for a Markovian process, the distinguishability between any two initial states of the open system, continuously diminishes over time. In other words, a quantum evolution, mathematically described by a quantum dynamical map $E_{t}$, is called Markovian if, for any arbitrary pair of initial quantum states $\rho^{1}\left(0\right)$ and $\rho^{2}\left(0\right)$, the evolved trace distance $D\left(\rho^{1}\left(t\right),\rho^{2}\left(t\right)\right)$ monotonically decreases with time. Hence, quantum Markovian dynamics exhibits a continuous loss of information from the open system to the environment. Consequently, a non-Markovian evolution is defined as a process in which, for certain time intervals, $dD\left(\rho^{1}\left(t\right),\rho^{2}\left(t\right)\right)/dt>0,$ usually interpreted as the information flowing back into the system temporarily. Provided that $E_{t}$ is invertible, one can show that the quantum process is BLP Markovian if and only if $E_{t}$ is P-divisible [3,73].

### 2.2. HSS-Based Witness of Non-Markovianity

Considering the distance measure [60]
(1)$${\left[d\left(p,q\right)\right]}^{2}=\frac{1}{2}\sum_{x}^{}{\left|p_{x}-q_{x}\right|}^{2},$$
where $p={\left\{p_{x}\right\}}_{x}$ and $q={\left\{q_{x}\right\}}_{x}$ denote the probability distributions, one can quantify the distance between infinitesimally close distributions taken from a one-parameter family $p_{x}\left(\phi \right)$ and then define the classical statistical speed as
(2)$$s\left[p(\phi_{0})\right]=\frac{d}{d\phi }d\left(p\left(\phi_{0}+\phi \right),p\left(\phi_{0}\right)\right).$$

These classical notions can be generalized to the quantum case by taking a pair of quantum states ρ and σ, and writing $p_{x}=\mathrm{Tr}\left\{E_{x}\rho \right\}$ and $q_{x}=\mathrm{Tr}\left\{E_{x}\sigma \right\}$ which represent the measurement probabilities corresponding to the positive-operator-valued measure (POVM) defined by $\left\{E_{x}\geq 0\right\}$ satisfying $\sum_{x}E_{x}=I$.

The associated quantum distance, which is called Hilbert–Schmidt distance [74], can be achieved by maximizing the classical distance over all possible choices of POVMs [75]
(3)$$D\left(\rho ,\sigma \right)\equiv \max_{\left\{E_{x}\right\}}d\left(p,q\right)=\sqrt{\frac{1}{2}\mathrm{Tr}\left[{\left(\rho -\sigma \right)}^{2}\right]}.$$

Consequently the HSS, i.e. the corresponding quantum statistical speed, is defined as follows:(4)$$HS\;S\left(\rho_{\phi }\right)\equiv HS\;S_{\phi }\equiv \max_{\left\{E_{x}\right\}}s\left[p(\phi )\right]=\sqrt{\frac{1}{2}\mathrm{Tr}\left[{\left(\frac{d\rho_{\phi }}{d\phi }\right)}^{2}\right]},$$
which can be easily computed without the diagonalization of $d\rho_{\phi }/d\phi$.

The recently proposed protocol, completely consistent with the BLP witness and used to detect non-Markovianity based on the HSS, is now briefly recalled [61]. We consider an *n*-dimensional quantum system whose initial state is given by
(5)$$|\psi_{0}\rangle =\frac{1}{\sqrt{n}}\left(e^{i\phi }|\psi_{1}\rangle +\ldots +|\psi_{n}\rangle \right),$$
where ϕ is an unknown phase shift and $\{|\psi_{1}\rangle ,\ldots ,|\psi_{n}\rangle \}$ denotes a complete and orthonormal set (basis) for the corresponding Hilbert space H. Given this initial state, the HSS-based witness of non-Markovianity is defined by
(6)$$\mathrm{Non}\text{-}\mathrm{Markovianity}\;\mathrm{Witness}:\chi \left(t\right)\equiv \frac{dHS\;S\left(\rho_{\phi }\left(t\right)\right)}{dt}>0,$$
in which $\rho_{\phi }\left(t\right)$ is the evolved state of the system.

### 2.3. Quantum Entanglement Measure

Quantum entanglement is a kind of quantum correlations which, from an operational point of view, can be defined as those correlations between different subsystems which cannot be generated by local operations and classical communication (LOCC) procedures. We use negativity [76] to quantify the quantum entanglement of the state, which is a reliable measure of entanglement in the case of qubit–qubit and qubit–qutrit systems [77].

For any bipartite state, $\rho_{AB}$, the negativity is defined as
(7)$$N\left(\rho_{AB}\right)=\sum_{i}\left|\lambda_{i}\right|,$$
where $\lambda_{i}$ is the negative eigenvalue of $\rho^{T_{k}}$, with $\rho^{T_{k}}$ denoting the partial transpose of the density matrix $\rho_{AB}$ with respect to subsystem k=A,B. Negativity can also be computed by the formula [78]
(8)$$N\left(\rho_{AB}\right)=\frac{1}{2}\left(∥\rho^{T_{k}}∥-1\right),$$
in which the trace norm of $\rho^{T_{k}}$ is equal to the sum of the absolute values of its eigenvalues [79], that is
(9)$$∥\rho^{T_{k}}∥=\sum_{i}\left|\mu_{i}\right|,$$
where the spectral decomposition of $\rho^{T_{k}}$ is given by $\sum_{i}\mu_{i}|i\rangle \langle i|$.

### 2.4. Quantum Correlation Quantifier: Measurement-Induced Disturbance

We use measurement-induced disturbance MID [59] as an alternative nonclassicality indicator for quantifying the quantum correlations of the bipartite quantum systems. It is defined as the minimum disturbance caused by local projective measurements leaving the reduced states invariant.

Considering the spectral resolutions of the reduced density states $\rho_{A}=\sum_{i}^{}p_{i}^{A}\Pi_{i}^{A}$ and $\rho_{B}=\sum_{j}^{}p_{j}^{B}\Pi_{i}^{B}$, one can compute the MID as follows:(10)$$M\left(\rho_{AB}\right)=I\rho_{AB}-I\left(\Pi \left(\rho_{AB}\right)\right),$$
where I is the mutual quantum information given by
(11)$$I\left(\rho_{AB}\right)=S\left(\rho_{A}\right)+S\left(\rho_{B}\right)-S\left(\rho_{AB}\right),$$
in which $S\left(\rho \right)=-\mathrm{tr}\rho \log\left(\rho \right)$ denotes the von Neumann entropy and
(12)$$\Pi \left(\rho_{AB}\right)=\sum_{i,j}\left(\Pi_{i}^{A}⊗\Pi_{j}^{B}\right)\rho_{AB}\left(\Pi_{i}^{A}⊗\Pi_{j}^{B}\right).$$

## 3. Analyzing the Efficiency of the HSS Witness in High-Dimensional Systems with Finite Hilbert Spaces

In this section, we check the sanity of HSS-based witness through several paradigmatic high-dimensional quantum systems with finite Hilbert spaces. The analyses are based on the fact that for systems in which the corresponding subsystems are coupled to independent environments, the oscillations of quantum correlations with time are associated with the non-Markovian evolution of the system [12,47,80], resulting in the transfer of correlations back and forth among the various parts of the total system. Moreover, by comparing the results presented in Refs. [10,61,81,82], we can demonstrate that the BLP measure of non-Markovianity can be used as a valid definition of non-Markovianity, when we intend to detect non-Markovianity by revivals of quantum correlations.

In particular, we consider a single qudit subject to a quantum environment, and a hybrid qubit-qutrit system coupled to independent as well as common quantum and classical noises. We show that the oscillation of the HSS-based witness is in qualitative agreement with nonmonotonic variations of the quantum resources, and hence it can be introduced as a faithful identifier of non-Markovianity in such high-dimensional systems with finite Hilbert spaces.

It should be noted that the efficiency of the HSS-based witness in detecting the non-Markovian nature of the dynamics directly depends on adopting the correct parametrization of the initial state of Equation (Equation 5), as discussed in Ref. [61]. However, often choosing the computational basis as the complete orthonormal set $\{|\psi_{1}\rangle ,\ldots ,|\psi_{n}\rangle \}$ is enough to capture the non-Markovianity, as shown in this paper. In all examples discussed below, the HSS is computed for the pure initial states while the quantum correlations may be calculated for mixed ones to illustrate the general efficiency off the HSS-based witness.

### 3.1. Single-Qudit Interacting with a Quantum Environment

#### 3.1.1. Coupling to a Thermal Reservoir

Let consider the spin-S systems interacting with a thermal reservoir modeled by an infinite chain of quantum harmonic oscillators with $\omega_{k}$, $b_{k}$, and $b_{k}^{†}$ being, respectively, the frequency, annihilation, and creation operators for the *k*-th oscillator. The total Hamiltonian of the system is given by
(13)$$H=\omega_{0}S_{z}+\sum_{k}\omega_{k}b_{k}^{†}b_{k}+\sum S_{z}\left(g_{k}b_{k}^{†}+g_{k}^{*}b_{k}\right),$$
in which $\omega_{0}$ denote the transition frequency between any neighboring energy states of the spin, and $S_{z}$, the *z* component of spin operator, can be represented by a diagonal matrix $S_{z}=\mathrm{diag}\left[s,s-1,\ldots ,-s\right]$ in the eigen-basis {|i〉,i=s,…,−s}. In the interaction picture Equation (Equation 13) into is expressed as
(14)$$H_{I}=\sum S_{z}\left(g_{k}b_{k}^{†}e^{i\omega_{k}t}+g_{k}^{*}b_{k}e^{-i\omega_{k}t}\right),$$
where $g_{k}$ denotes the coupling strength between the spin and the environment through the dephasing interaction. Up to an overall phase factor, the corresponding unitary propagator is obtained as
(15)$$V\left(t\right)=\exp\left[\frac{1}{2}S_{z}\sum_{k}\left(\alpha_{k}b_{k}^{†}-\alpha^{*}b_{k}\right)\right],$$
where $\alpha_{k}=2g_{k}\left(1-e^{i\omega_{k}t}\right)/\omega_{k}$.

It is assumed that the initial state of the spin-bath system is in a product state $\rho_{T}\left(0\right)=\rho \left(0\right)⊗\rho_{B}$ in which $\rho \left(0\right)$ denotes the initial state of spin, and
(16)$$\rho_{B}=\frac{1}{Z_{B}}e^{-\beta \sum_{k}\omega_{K}b_{k}^{†}b_{k}}$$
represents the thermal equilibrium state of the bath with partition function $Z_{B}$ and inverse temperature $\beta =\frac{1}{k_{B}T}.$ The evolved state of the system can be calculated by [83]
(17)$$\rho_{nm}\left(t\right)=\rho_{nm}\left(0\right)\exp\left[-{\left(n-m\right)}^{2}\Gamma \left(t\right)\right],$$
where n,m=−s,−s+1,…,0,…,s−1,s and, in the continuum-mode limit, the decoherence function is given by
(18)$$\Gamma \left(t\right)=\int_{0}^{\infty }J\left(\omega \right)\coth\left(\frac{\omega }{2k_{b}T}\right)\frac{1-\cos(\omega t)}{\omega^{2}}d\omega ,$$
with spectral density $J\left(\omega \right)=\sum_{k}{\left|g_{k}\right|}^{2}\delta \left(\omega -\omega_{k}\right)$.

The Γ(t) behavior closely depends on the characteristics of the environment. Here we consider the Ohmic-like reservoirs with spectral density
(19)$$J\left(\omega \right)=\alpha \frac{\omega^{s}}{\omega_{c}^{s-1}}\exp\left(\frac{-\omega }{\omega_{c}}\right),$$
where α represents a dimensionless coupling strength, and $\omega_{c}$ denotes the cutoff frequency of the bath. Changing the Ohmic parameter *s*, one can obtain sub-Ohmic (0<s<1), Ohmic (s=1) and super-Ohmic (s>1) reservoirs.

#### 3.1.2. Coupling to a Squeezed Vacuum Reservoir

In the case that the spin system is coupled to a squeezed vacuum reservoir, the reduced density-matrix elements are similar to the ones presented in Equation (Equation 17) when the decoherence function Γ(t) is replaced by
(20)$$\begin{array}{cc} \gamma \left(t\right) & =\int_{0}^{\infty }J\left(\omega \right)\frac{\left(1-\cos\left(\omega t\right)\right)}{\omega^{2}}\left[\cosh(2r)-\sinh(2r)\cos\left(\omega t-\theta \right)\right]d\omega , \end{array}$$
where *r* is the squeezed amplitude parameter, and θ denotes the squeezed angle.

Because the structures of the density matrices are the same in both scenarios (coupling to thermal and squeezed vacuum reservoirs), we only focus on the interaction of the system with the squeezed vacuum reservoir, noting that the general results also holds for the thermal reservoir.

We take the qudit in the pure initial state
(21)$$|\psi \rangle =\frac{1}{\sqrt{2s+1}}(e^{i\phi }\left|s\rangle +|s-1\rangle +|s-2\rangle +\cdots +|-s\rangle \right),$$
which leads to the evolved state $\rho \left(t\right)$ given by
(22)$$\rho \left(t\right)=\frac{1}{2s+1}\left(\begin{array}{cccc} 1 & e^{-\gamma \left(t\right)}e^{i\phi } & \cdots & e^{-{\left(2s\right)}^{2}\gamma \left(t\right)}e^{i\phi }\\ e^{-\gamma \left(t\right)}e^{-i\phi } & 1 & \cdots & e^{-{\left(2s-1\right)}^{2}\gamma \left(t\right)}\\ e^{-4\gamma \left(t\right)}e^{-i\phi } & e^{-\gamma \left(t\right)} & \cdots & e^{-{\left(2s-2\right)}^{2}\gamma \left(t\right)}\\ \vdots & 1 & \ddots \\ e^{-{\left(2s\right)}^{2}\gamma \left(t\right)}e^{-i\phi } & e^{-{\left(2s-1\right)}^{2}\gamma \left(t\right)} & \cdots & 1 \end{array}\right).$$

Therefore, the time derivative of the HSS-based witness is obtained as
(23)$$\chi \left(t\right)=-\frac{1}{2s+1}\frac{\partial \gamma \left(t\right)}{\partial t}\frac{\sum_{k=1}^{2s}k^{2}e^{-2k^{2}\gamma \left(t\right)}}{\sum_{k=1}^{2s}e^{-2k^{2}\gamma \left(t\right)}}.$$

The HSS-based witness χ(t)>0 tells us that the process is non-Markovian whenever $\frac{\partial \gamma \left(t\right)}{\partial t}<0$, which corresponds to time intervals in which the decoherence function decreases, leading to the re-coherence phenomenon. As known, in this system the non-Markovian effects, originating from the non-divisible maps, appear when the decoherence function temporarily decays with time [84]. Therefore, our witness correctly predicts the intervals at which the memory effects arise in this single-qudit system. Moreover, when γ(t) is a monotonous increasing function of time, the dynamics is Markovian because the coherence decays monotonously with time.

### 3.2. Hybrid Qubit–Qutrit System Interacting with Various Quantum and Classical Environments

The composite hybrid qubit(*A*)–qutrit(*B*) system consists of a spin–$\frac{1}{2}$ subsystem (qubit A) and a spin-1 subsystem (qutrit B). In the following, we study the interaction of this composite system with local non-Markovian environments *A* and *B*, or with a common environment *C* modeling quantum or classical noises. The theoretical schematic of this system is depicted in Figure 1.

#### 3.2.1. Coupling to Independent Squeezed Vacuum Reservoirs

Now we investigate the scenario in which each of the subsystems, i.e., the qubit *A*$\left(s_{A}=\frac{1}{2}\right)$ and qutrit *B*$\left(s_{B}=1\right)$, interacts independently with its local squeezed vacuum reservoir. For simplicity we assume that the characteristics of the reservoirs are similar. Equation (Equation 17), with the decoherence factor introduced in Equation (Equation 20), gives the reduced density matrices of the subsystems. Computing them and applying the method presented in [81], one can obtain the elements of the evolved density matrix of the composite system as [85]
(24)$${\rho_{AB}}_{nm}\left(t\right)={\rho_{AB}}_{nm}\left(0\right)\exp[-{\left(n_{A}-m_{A}\right)}^{2}-{\left(n_{B}-m_{B}\right)}^{2}]\gamma \left(t\right),$$
where $n_{A},m_{A}=-s_{A},\ldots ,s_{A}$ and $n_{B},m_{B}=-s_{B},\ldots ,s_{B}$.

**Pure initial state.** We take the hybrid qubit–qutrit system initially in a pure state given by [61]
(25)$$|\psi \rangle =\frac{1}{\sqrt{6}}\left(e^{i\phi }|00\rangle +|01\rangle +|02\rangle +|10\rangle +|11\rangle +|12\rangle \right),$$
which leads to a dynamics of the system described by the evolved reduced density matrix $\rho \left(t\right)$ whose elements are presented in Section A.1. Then, the HSS is obtained as
(26)$$HS\;S=\frac{1}{6}\sqrt{2e^{-2\gamma \left(t\right)}+e^{-4\gamma \left(t\right)}+e^{-8\gamma \left(t\right)}+e^{-10\gamma \left(t\right)}}.$$

The dynamics of negativity, MID and HSS computed by the evolved state of the system are plotted in Figure 2. We find that each of the measures initially decreases with time, then starts to increase, and finally remains approximately constant over time, a behavior known as the freezing phenomenon [86,87,88,89,90,91,92]. As discussed, the revival of the quantum correlation measures can be attributed to the non-Markovian evolution of the system [47]. We see that the behaviors of the HSS, negativity and quantum correlation exhibit an excellent qualitative agreement. Consequently, the HSS-based witness can precisely capture the non-Markovian dynamics of the composite system.

**Mixed initial state.** The non-Markovianity of the system, as faithfully individuated by quantum correlation measures, may in general depend on the initial state. It is thus important to investigate whether the HSS witness, obtained from the initial pure state of Equation (Equation 25) by definition, is capable to identify the non-Markovian character of the system dynamics also when the system starts from a mixed state. We shall study this aspect here and in all the other environmental conditions considered hereafter (see sections below devoted to a mixed initial state).

We consider the one-parameter mixed entangled state as the initial state of the hybrid qubit–qutrit system [93]
(27)$$\rho_{0}\left(p\right)=\frac{p}{2}\left(|01\rangle \langle 01|+|11\rangle \langle 11|\right)+p|\psi^{+}\rangle \langle \psi^{+}|+\left(1-2p\right)|\psi^{-}\rangle \langle \psi^{-}|,$$
where
(28)$$\begin{array}{c} |\psi^{+}\rangle =\frac{1}{\sqrt{2}}\left(|00\rangle +|12\rangle \right),\\ |\psi^{-}\rangle =\frac{1}{\sqrt{2}}\left(|02\rangle +|10\rangle \right), \end{array}$$
in which the entanglement parameter *p* varies from 0 to 1 such that ρ(p) is entangled except for $p=\frac{1}{3}$. We point out that such a state is taken as the initial state of the system for the dynamics of the quantum correlation quantifiers, namely negativity and MID. We find that Equation (Equation 27) leads to the evolved state of the system
(29)$$\rho \left(t\right)=\left(\begin{array}{cccccc} \frac{p}{2} & 0 & 0 & 0 & 0 & \frac{p}{2}F\\ 0 & \frac{p}{2} & 0 & 0 & 0 & 0\\ 0 & 0 & \frac{1-2p}{2} & \frac{1-2p}{2}F & 0 & 0\\ 0 & 0 & \frac{1-2p}{2}F & \frac{1-2p}{2} & 0 & 0\\ 0 & 0 & 0 & 0 & \frac{p}{2} & 0\\ \frac{p}{2}F & 0 & 0 & 0 & 0 & \frac{p}{2} \end{array}\right),$$
where $F=e^{-5\gamma \left(t\right)}$. Then, the negativity is given by [71]
(30)$$\begin{array}{cc} N= & \frac{\left(p-1\right)}{2}+\frac{1}{4}\left|p+(1-p)F\right|+\frac{1}{4}\left|p-(1-p)F\right|+\frac{1}{4}\left|p-(1-2p)F\right|+\\ & \frac{1}{4}\left|p+(1-2p)F\right|. \end{array}$$
Moreover, using Equation (Equation 10) we can compute the MID as
(31)$$M=\frac{\left(1-p\right)}{2}\left[\left(1+F\right)\log\left(1+F\right)+\left(1-F\right)\log\left(1-F\right)\right].$$

In Figure 3, we compare the evolution of HSS, obtained from the initial pure state of Equation (Equation 25), with the dynamics of negativity and MID, computed for the mixed initial state of Equation (Equation 15), for different values of *p*. The dynamics of the HSS is again in perfect agreement with that observed for the entanglement and quantum correlations as quantified by the negativity and MID, respectively. Therefore, the HSS-based witness, computed versus the phase parameter encoded into an initial pure state of the system, can efficiently detect the non-Markovian dynamics even in the case when the initial state of our high-dimensional system is not pure. It should be noted that in the presence of sudden death of entanglement, which occurs for some values of the entanglement parameter (for example, for p=0.4), only the HSS and MID show the same dynamics. Hence, the negativity cannot be used as a faithful witness of non-Markovianity when it exhibits the sudden death phenomenon.

In the case of initially entangled noninteracting qubits in independent non-Markovian quantum environments, entanglement or quantum correlation revivals can be explained in terms of transfer of correlations back and forth from the composite system to the various parts of the total system. This is due to the back-action via the environment on the system, which creates correlations between qubits and environments and between the environments themselves. Accordingly, in this case the non-Markovianity is defined as backflow of information from the environment(s) to the system(s).

#### 3.2.2. Coupling to Classical Environments

Here we assume that the hybrid qubit-qutrit system is affected by a classical environment implemented by random telegraph noise (RTN) with a Lorentzian spectrum. It is a famous class of non-Gaussian noises used to generate the low-frequency $\frac{1}{f^{\alpha }}$ noise both theoretically and experimentally. It is also responsible for coherent dynamics in quantum solid-state nanodevices [94,95,96]. Physically, the RTN may result from one of the following scenarios: (i) charges flipping between two locations in space (charge noise); (ii) electrons trapping in shallow subgap formed at the boundary between a superconductor and an insulator (noise of critical current); and (iii) spin diffusion on a superconductor surface generated by the exchange mediated by the conduction electrons (flux noise) [97,98]. The Hamiltonian of the qubit–qutrit system under the RTN is given by
(32)$$\begin{array}{cc} H\left(t\right)= & H_{0}+H_{I}\\ H_{0}= & \sum_{k=A,B}\epsilon_{k}S_{k}^{Z},H_{I}=\sum_{k=A,B}\left[J_{k}L_{k}\left(t\right)+J_{c}C\left(t\right)\right]S_{z}^{k}, \end{array}$$
where $\epsilon_{k}$ denote the energy of an isolated qubit (qutrit), $S_{z}^{A}=\sigma_{z}$ and $S_{z}^{B}$ represent the spin operators of, respectively, the qubit and the qutrit in the *z*-direction. Moreover, $J_{k}$ and $J_{c}$ represent the coupling strengths of each marginal system to the local and non-local RTN, such that we consider two types of system-environment interactions, namely
(1)Local or independent environments (ie): $J_{k}=\nu \neq 0$ and $J_{c}=0$;(2)Non-local or common environments (ce): $J_{k}=0$ and $J_{c}=\nu \neq 0$.

Furthermore, $L_{k}\left(t\right)$ and C(t) denote the random variables used to introduce the stochastic processes. They are used to describe the different conditions under which the subsystems undergo decoherence due to the environment. Here, they represent classical random fluctuating fields such as bistable fluctuators flipping between two fixed values ±m at rates $\gamma_{k}$ and γ, respectively. For simplicity, we assume that $\gamma_{k}=\gamma$. For the *autocorrelation function* of the random variable $\eta \left(t\right)=\left\{L_{k}\left(t\right);C\left(t\right)\right\}$ we have $\langle \delta \eta \left(t\right)\delta \eta \left(t^{\prime }\right)\rangle =\exp\left[-2\gamma |t-t^{\prime }|\right]$ with a Lorentzian power spectrum $S\left(\omega \right)=\frac{4\gamma }{\omega^{2}+\gamma^{2}}$. Defining the parameter $q=\frac{\gamma }{\nu }$, we can identify two regimes for the dynamics of quantum correlations: the Markovian regime (q≫1: fast RTN), and the non-Markovian regime (q≪1: slow RTN). The time-evolving state of the system under the influence of the RTN is given by
(33)$$\rho \left(\{\eta \},t\right)=U\left(\{\eta \},t\right)\rho \left(0\right)U^{†}\left(\{\eta \},t\right).$$
in which the time-evolution operator $U\left(\{\eta \},t\right)$ called the stochastic unitary operator in the interaction picture is given by
(34)$$U\left(\{\eta \},t\right)=\exp\left[-i\int_{0}^{t}H_{I}\left(t^{\prime }\right)dt^{\prime }\right].$$
where $\eta \left(t\right)=\left\{L_{k}\left(t\right);C\left(t\right)\right\}$ stands for the different realizations of the stochastic process. Because $U\left(\{\eta \},t\right)$ depends on the noise, we should perform the ensemble average over the noise fields to obtain the reduced density matrix of the open system, i.e.,
(35)$$\rho_{ie(ce)}={\langle \rho \left(\{\eta \},t\right)\rangle }_{\eta \left(t\right)}.$$
The evolved state of the system in the presence of independent environments (ie) and collective environments (ce) is obtained as
(36)$$\begin{array}{c} \rho_{ie}\left(t\right)={\langle {\langle \rho \left(\theta_{A}\left(t\right),\theta_{B}\left(t\right),t\right)\rangle }_{\theta_{A}}\rangle }_{\theta_{B}}\\ \rho_{ce}\left(t\right)={\langle \rho \left(\theta \left(t\right),t\right)\rangle }_{\theta }, \end{array}$$
where $\theta_{k}\left(t\right)=\nu \int_{0}^{t}L_{k}\left(t^{\prime }\right)dt^{\prime }$ (k=A,B) and $\theta \left(t\right)=\nu \int_{0}^{t}C\left(t^{\prime }\right)dt$. Calculation of the above terms requires the computation of averaged terms of the type $\langle e^{\pm in\theta }\rangle$$\left(n\in N\right)$ given by [99]
(37)$$\begin{array}{c} \langle e^{in\theta }\rangle =D_{n}\left(\tau \right)=\langle \cos\left(n\theta \right)\rangle \pm i\langle \sin\left(n\theta \right)\rangle ,\\ \langle \sin\left(n\theta \right)\rangle =0, \end{array}$$
$$\langle \cos\left(n\theta \right)\rangle =\left\{\begin{array}{cc} e^{-q\tau }\left[\cosh\left(\xi_{qn}\tau \right)+\frac{q}{\xi_{qn}}\sinh\left(\xi_{qn}\tau \right)\right], & \;q>n\\ e^{-q\tau }\left[\cos\left(\xi_{nq}\tau \right)+\frac{q}{\xi_{nq}}\sin\left(\xi_{nq}\tau \right)\right], & q\;<n \end{array}\right.$$
where $\xi_{ab}=\sqrt{a^{2}-b^{2}}$((a,b)=n,q), and τ=νt denotes the scaled (dimensionless) time [71].

**Pure initial state in the presence of independent classical environments.** Here, we assume that each of the qubits and qutrits interact locally with local RTN, while the composite system starts with the pure initial state in Equation (Equation 25). For this case, the elements of evolved density matrix are given in Section A.2. Then the HSS is obtained as
(38)$$\begin{array}{c} HS\;S=\frac{1}{6}\sqrt{D_{1}^{2}\left(\tau \right)+2D_{2}^{2}\left(\tau \right)+D_{2}^{2}\left(\tau \right)D_{1}^{2}\left(\tau \right)+D_{2}^{4}\left(\tau \right)}. \end{array}$$

In Figure 4, we illustrate the time behaviors of the negativity, MID and HSS in the non-Markovian regime as a function of the dimensionless time. It is clear that when the entanglement sudden death occurs, the HSS and MID synchronously oscillate with time as they are suppressed to the minimum value and then rise. Moreover, at the first revival of the measures, the minimum point of the HSS exactly coincides with that of the negativity. After that moment we see that maximum (minimum) points of the HSS are in complete coincidence with maximum (minimum) points of the negativity as well as the MID. This perfect qualitative agreement between HSS and entanglement or quantum correlations is evidence that the HSS-based witness can precisely detect non-Markovianity in the presence of classical noises.

**Mixed initial state in the presence of independent classical environments.** Now we compare the dynamics of the HSS, obtained from the initial pure state of Equation (Equation 25), with the evolution of the negativity and quantum correlation computed for the initial mixed state of Equation (Equation 27). The evolved density matrix, the corresponding negativity and quantum correlation are obtained from, respectively, Equations (Equation 29)–(Equation 31) replacing F with ${D_{2}\left(\tau \right)}^{2}$.

Figure 5 exhibits this comparison for different values of the entanglement parameter *p*. Not considering the periods when the sudden death of the entanglement occurs, we observe that the maximum and minimum points of the measures are very close to each other and small deviations originate from the fact that the initial state, used for computation of the HSS-based measure, should be optimized over all possible parametrizations. Therefore, the HSS-based measure remains as a valid non-Markovianity identifier in the presence of the classical noises.

**Mixed initial state in the presence of a common classical environment.** Let us now compare the dynamics of the HSS, obtained as usual from the initial pure state of Equation (Equation 25) by definition, with the evolution of the negativity and quantum correlation computed for the initial mixed state of Equation (Equation 27), when both the qubit and the qutrit are embedded into a common RTN source in the non-Markovian regime. The elements of the evolved dynamical density matrix are given in Section A.3. Then, one can easily determine the HSS as
(39)$$HSS=\frac{1}{6}\sqrt{{D_{1}\left(\tau \right)}^{2}+2{D_{2}\left(\tau \right)}^{2}+{D_{3}\left(\tau \right)}^{2}+{D_{4}\left(\tau \right)}^{2}}.$$

Moreover, the evolved density matrix of the hybrid qubit–qutrit system for the initial mixed state of Equation (Equation 27) is obtained as
(40)$$\rho \left(t\right)=\left(\begin{array}{cccccc} \frac{p}{2} & 0 & 0 & 0 & 0 & \frac{p}{2}Fe^{i\phi }\\ 0 & \frac{p}{2} & 0 & 0 & 0 & 0\\ 0 & 0 & \frac{1-2p}{2} & \frac{1-2p}{2} & 0 & 0\\ 0 & 0 & \frac{1-2p}{2} & \frac{1-2p}{2} & 0 & 0\\ 0 & 0 & 0 & 0 & \frac{p}{2} & 0\\ \frac{p}{2}Fe^{-i\phi } & 0 & 0 & 0 & 0 & \frac{p}{2} \end{array}\right),$$
where $F=D_{4}\left(\tau \right)$.

As a consequence, we find that the negativity and MID are, respectively,
(41)$$\begin{array}{cc} N= & \frac{1}{4}\left[(p-1)+|3p-1|+|(1-2p)-pF|+|(1-2p)+pF|\right], \end{array}$$
(42)$$\begin{array}{cc} M= & \left(1-2p\right)+\frac{p}{2}\left(1+F\right)\log\left(1+F\right)+\frac{p}{2}\left(1-F\right)\log\left(1-F\right). \end{array}$$

For common environments, we know that mutual interaction between subsystems, induced by the common environment, may lead to the preservation of correlations or even result in creation of quantum correlations between the subsystems [82,100,101,102]. Therefore, revivals of the quantum correlations cannot be necessarily linked to pure non-Markovianity effects and hence we do not expect complete consistency between the HSS and quantum correlations behaviors (see Figure 6 demonstrating this feature of common environments causing the MID to fail in detecting non-Markovianity). Except for these situations, we see that the maximum (minimum) points of the HSS computed for the initial pure state are very close to those of the MID calculated for the initial mixed state.

It should be noted that the classical environments cannot store any quantum correlations on their own, and hence they do not become entangled with their respective quantum systems. Accordingly, common interpretation of non-Markovianity in accordance with inflow (outflow) of information to (from) the system may be problematic in the presence of the RTN and other similar classical noises [47,103]. In other words, it is somewhat misleading to talk about information flow from the system(s) to the environment(s) or information backflow from the environment(s) to the system(s). The better interpretation is to say that the quantum system has a recording memory of the events affecting its dynamics. When the quantum memory starts remembering, the information about the past events becomes accessible, leading to revival of the quantum correlations and hence to the appearance of quantum non-Markovianity [104].

#### 3.2.3. Composite Classical-Quantum Environments

Here we investigate a hybrid system formed by a qubit subjected to a random telegraph noise and a qutrit independently subjected to a squeezed vacuum reservoir. The Hamiltonian of such a system can be written as
(43)$$H=H_{\mathrm{qb}}\left(t\right)⊗I_{\mathrm{qt}}+I_{\mathrm{qb}}⊗H_{\mathrm{qt}}\left(t\right).$$
where $I_{\mathrm{qb}}\left(\mathrm{qt}\right)$ denotes the identity operator acting on the subspace of the qubit (qutrit). Moreover, the Hamiltonians of the local interaction of the qubit and qutrit, $H_{qb}\left(t\right)$ and $H_{qt}\left(t\right)$, as well as their corresponding evolution operators, $U_{qb}\left(\theta ,t\right)$ and $U_{qt}\left(\theta ,t\right)$ can be extracted from Section 3.2.2 and Section 3.1. In addition, one can consider the unitary evolution operator of the system as $U=U_{qb}\left(\theta ,t\right)⊗U_{qt}\left(t\right).$ Then, the evolved density matrix of the this system can then be obtained by averaging the unitary evolved density matrix over the stochastic process induced by the RTN.

**Pure initial state.** The elements of the evolved density matrix when starting from the pure state of Equation (Equation 25) are given in Section A.4, leading to the following expression for the HSS:(44)$$HSS=\frac{1}{6}\sqrt{\left(e^{-2\gamma \left(t\right)}+e^{-8\gamma \left(t\right)}\right)\left(1+D_{2}{\left(\tau \right)}^{2}\right)+D_{2}{\left(\tau \right)}^{2}}.$$

The time behaviors of negativity, MID and HSS are shown in Figure 7 illustrating that all measures exhibit simultaneous oscillations with time such that their maximum and minimum points exactly coincide. This excellent agreement confirms the faithfulness of the HSS-based measure to detect memory effects.

**Mixed initial state.** Using Equation (Equation 27) as the initial state and computing the evolved state of the system (see Section B.4), we find that the the negativity and MID, respectively, are in the form of Equations (Equation 30) and (Equation 31) with $F=D_{2}\left(\tau \right)e^{-4\gamma (t)}$. In Figure 8, the dynamics of negativity and MID, obtained for the initial mixed state, has been compared with that of the HSS (computed for the initial pure state) in the non-Markovian regime.

The related analyses are similar to those in the above discussed scenarios, showing that the HSS-based witness may be a proper non-Markovianity identifier even if the initial state of high-dimensional systems is not pure.

## 4. Conclusions

Recently, the HSS-based witness, a quantifier of quantum statistical speed which has the advantage of avoiding the diagonalization of the evolved density matrix, has been introduced as a trustful witness of non-Markovianity in low-dimensional systems [61]. In this work, we have generalized this result showing that the proposed witness is a bona-fide identifier of non-Markovianity for high-dimensional and multipartite open quantum systems with finite Hilbert spaces. This result stems from the observation that the HSS-based witness is in perfect agreement with established non-Markovianity identifiers based on the dynamical breakdown of monotonicity for quantum information resources, such as negativity and measurement-induced disturbance. We have found that, despite the common interpretation of non-Markovianity in terms of backflow of information from the environment to the system may be problematic [6], the HSS-based witness is capable to detect memory effects of the evolved quantum system.

In order to construct a non-Markovianity measure on the basis of a geometric distance between two quantum states, one of desirable properties is that the distance is contractive, i.e., nonincreasing under any completely positive trace preserving (CPTP) map. It has been shown that the HSS is contractive under CPTP maps in low-dimensional Hermitian systems [61]. Checking all of the dynamical cases presented here, we have found that the contractivity of the HSS holds not only in low dimensional systems but also in finite high-dimensional ones. Recently, an HSS-like measure has been used to analyze the quantum speed limit for continuous-variable systems following Gaussian preserving dynamics [105]. Therefore, our results also motivate further studies about HSS applications in detecting non-Markovianity in continuous variable systems.

By definition, the HSS-based witness of memory effects is obtained by maximizing the speed of a classical distance measure between the probability distributions, over all quantum measurements. This, as a prospect, may induce the idea of the the possibility to use classical-like description of density matrix properties in probability representation of quantum mechanics.

Recently, K. Goswami et al. [106] have reported a quantum-optics experimental setup to implement a non-Markovian process—specifically, a process with initial classical correlations between system and environment. It should be noted that in all systems investigated in this paper we have adopted the usual assumption that the system and its environment are initially uncorrelated. It would be interesting to generalize the application of the HSS-based non-Markovianity witness to scenarios in which initial correlations between the system and environment rise. This will be studied in detail in our future work.

## Figures and Tables

**Figure 1 entropy-24-00395-f001:**
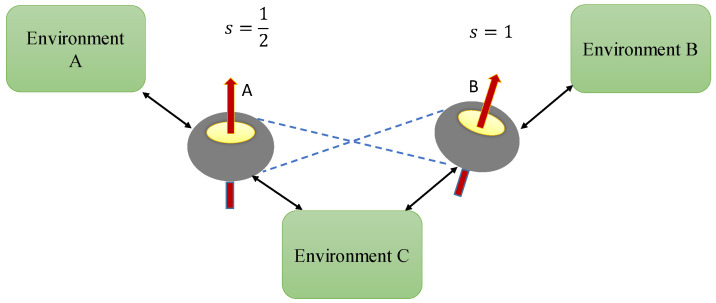
Illustration of the composite qubit(*A*)-qutrit(*B*) system; Blue dashed lines represent entanglement between the subsystems. The bipartite system can interact either with independent local environments $E_{A}$, $E_{B}$ or with a common environment $E_{C}$.

**Figure 2 entropy-24-00395-f002:**
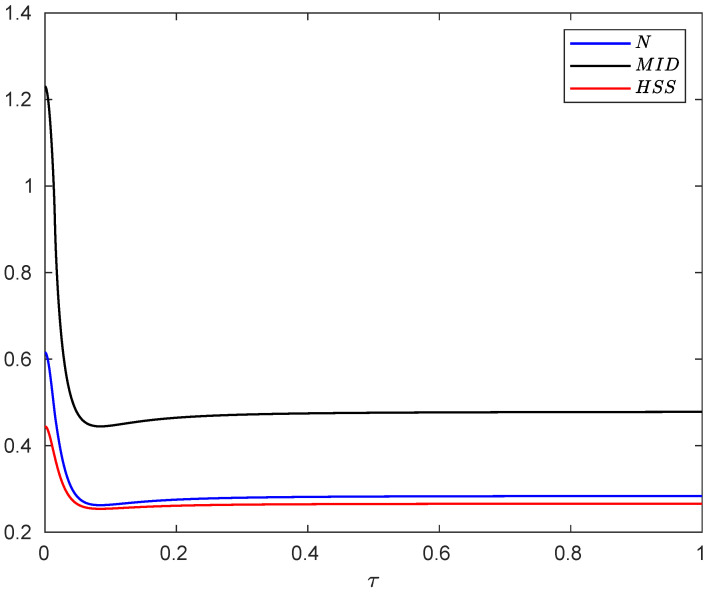
Evolution of the negativity, MID and HSS as a function of dimensionless time $\tau =\omega_{0}t$ when each subsystem of the hybrid qubit–qutrit system, starting from the initial pure state, is independently subject to a squeezed vacuum reservoir. The values of the other parameters are α=0.1, $\omega_{c}=20\omega_{0}$, r=0.3, ϕ=π and s=3.

**Figure 3 entropy-24-00395-f003:**
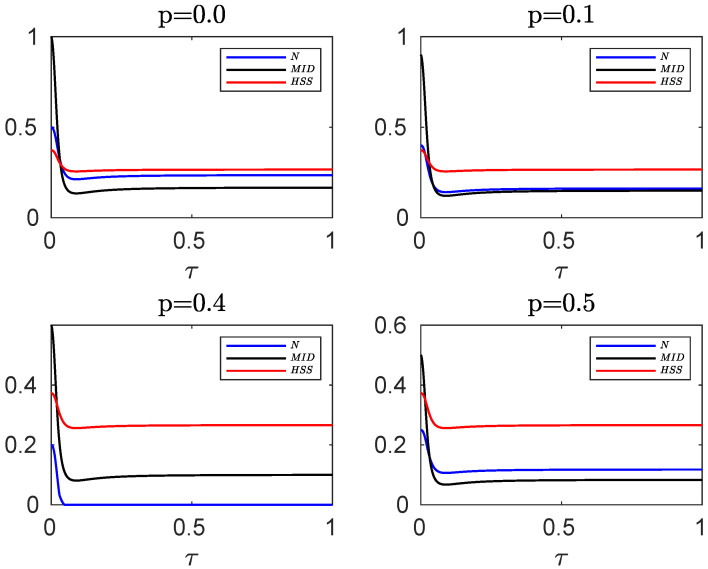
Comparing the evolution of negativity and MID computed for the initial mixed state of the hybrid qubit–qutrit system, when each subsystem is independently coupled to a squeezed vacuum reservoir, with HSS (obtained from the initial pure state) for different values of the entanglement parameter *p*. In all plots the remaining parameters are α=0.1, s=3, $\omega_{c}=20\;\omega_{0}$, r=0.3.

**Figure 4 entropy-24-00395-f004:**
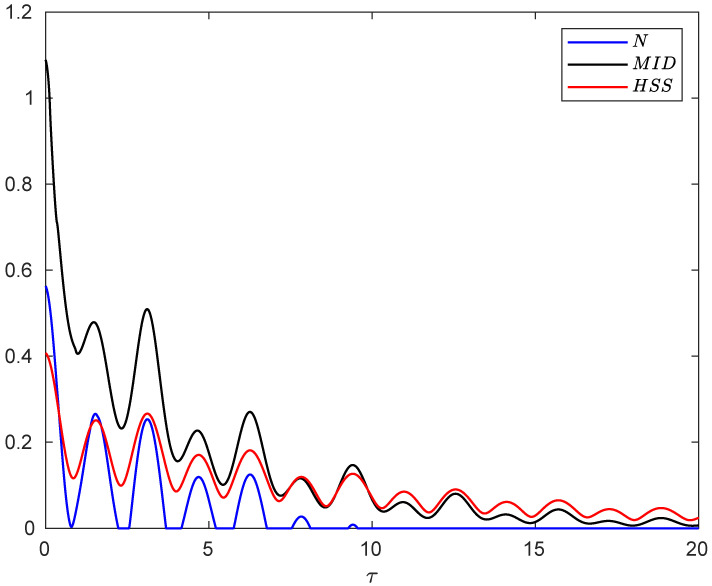
Evolution of negativity, MID and HSS as a function of dimensionless time τ=νt when each subsystem of the hybrid qubit–qutrit system, starting from the initial pure state, is independently subject to a random telegraph noise in non-Markovian regime q=0.1.

**Figure 5 entropy-24-00395-f005:**
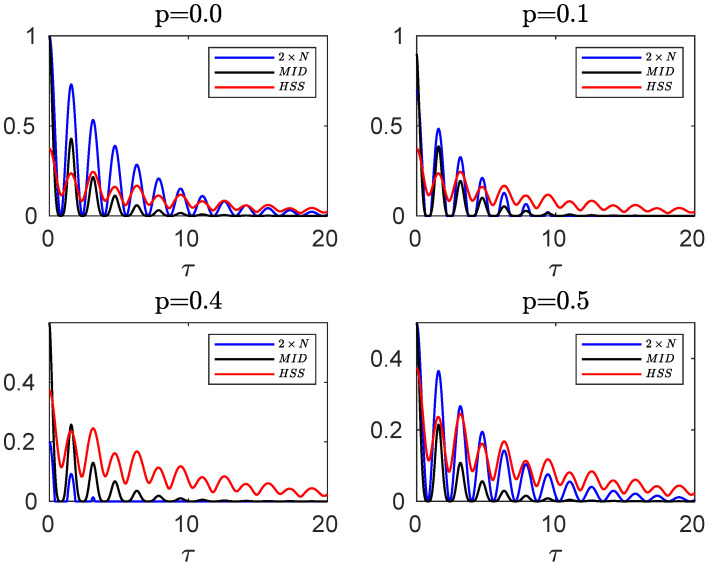
Comparing the evolution of negativity and MID computed for the initial mixed state of the hybrid qubit–qutrit system, when each subsystem is independently coupled to a random telegraph noise, with HSS (obtained from the initial pure state) for different values of the entanglement parameter *p* in the non-Markovian regime: q=0.1.

**Figure 6 entropy-24-00395-f006:**
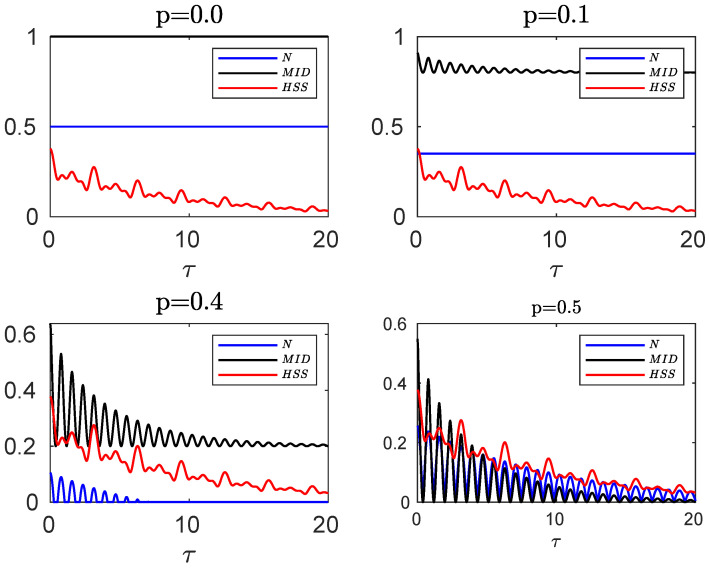
Comparing the evolution of negativity and MID computed for the initial mixed state of the hybrid qubit–qutrit system, when its subsystems are subject to a common RTN source, with HSS (obtained from the initial pure state) for different values of the entanglement parameter *p* in the non-Markovian regime: q=0.1.

**Figure 7 entropy-24-00395-f007:**
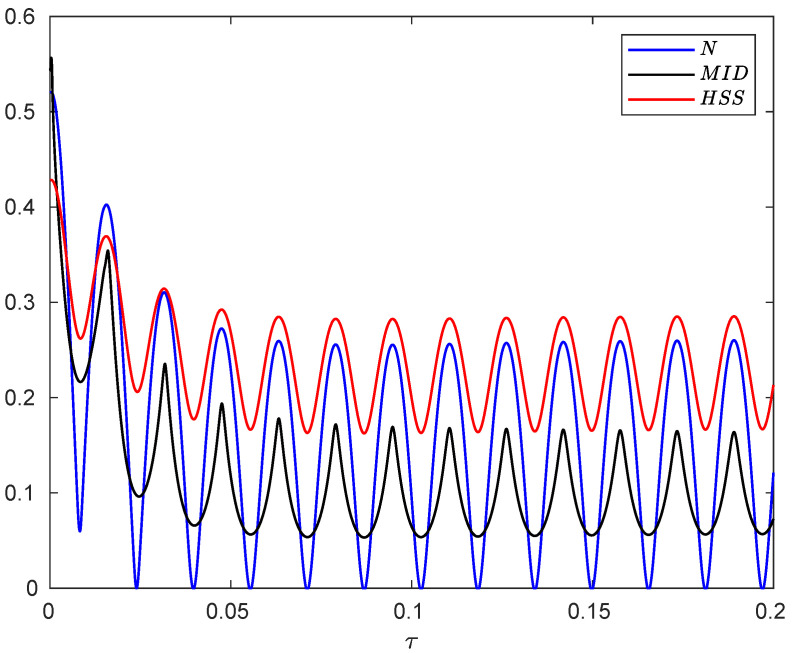
Evolution of negativity, MID and HSS as a function of dimensionless time τ when the subsystems of the hybrid qubit–qutrit system, starting from the initial pure state, are independently subject to composite classical-quantum environments. The values of the other parameters are given by α=0.1, $\omega_{c}=20\;\omega_{0}$, r=0.3, and ν=100.

**Figure 8 entropy-24-00395-f008:**
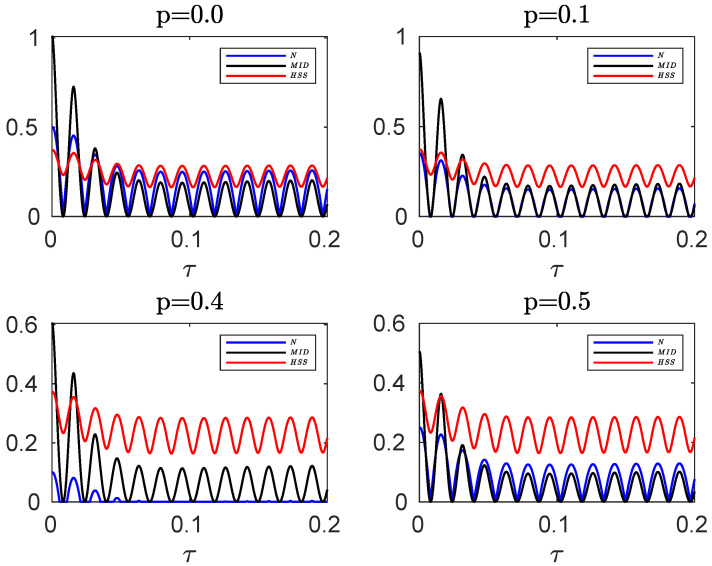
Comparing the evolution of the negativity and MID, computed for the initial mixed state of the hybrid qubit–qutrit system, when the subsystems are independently subject to composite classical-quantum environments, with the HSS obtained from the initial pure state for different values of the entanglement parameter *p* in the non-Markovian regime: q=0.1. The values of the other parameters are given by α=0.1, s=3, $\omega_{c}=20\;\omega_{0}$, p=0 and v=100.

## Data Availability

Not applicable.

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
