# Peer review of "Memory Effects in High-Dimensional Systems Faithfully Identified by Hilbert–Schmidt Speed-Based Witness"

_entropy, 2022, doi:10.3390/e24030395_

Round 1

Reviewer 1 Report

This manuscript presents a study of the Hilbert Schmidt speed (HSS) as a witness of non-Markovian behavior in  higher-dimensional and multipartite open quantum systems under different scenarios.  This study goes beyond the use of HSS in low dimensional systems and addresses the use of HSS as a non-Markovian witness in higher dimensions. The authors compare the time behavior of HSS with that of negativity and MID to illustrate the similarity in the behaviors.

The manuscript is well-written and well-presented. I have a few minor comments concerning the presentation of the figures.

In the discussion of Fig. 2 at the end of p. 6, what is the freezing phenomenon that the authors invite the reader to appreciate? That the curves approach a constant value for larger values of tau? This could be made clearer.

In the discussion of Fig. 2, an important point is that non-Markovian behavior is illustrated by an increase in the values of the curves (revivals). However, it is difficult for the reader to appreciate these increases, especially for the N and HSS measures. Would this behavior be more apparent if each curve were presented separately with its own scaling? I also had this difficulty with Fig. 3.

In Fig. 4, in the first revival, it seems that MID does not exactly coincide with N and HSS, while it seems to for larger tau? Is there any reason for this? Check ‘non-Markovian’ in the figure caption.

In Fig. 5, it is difficult to discern the blue N curve for p=0.1. See also Fig. 8.

In Fig. 6, the reader is left with the impression that there is less structure in the HSS curves (red) as compared to the MID ones, for p=0.1, p=0.4 and p=0.5. Is this so, or due to the presentation? Would making all curves solid (instead of dash dot) aid in the presentation? The colors would distinguish the curves. The frame around the p=0.5 plot is boldfaced (also the inset with legend) and is not consistent with the remainder. See also Figs. 3, 5, 8.

Check 'nonmonotonic' on line 31. 

Reference [59] is duplicated in [69].

See also [48] and [93].

Author Response

Reply to Reviewer 1

We would like to thank the referee for the constructive and useful comments which contributed to improve the manuscript. In the following we give our replies to all the raised points:

0) Referee: The manuscript is well-written and well-presented. I have a few minor comments concerning the presentation of the figures.

Our reply: We thank the referee for positive assessment of our work.

1) Referee: In the discussion of Fig. 2 at the end of p. 6, what is the freezing phenomenon that the authors invite the reader to appreciate? That the curves approach a constant value for larger values of tau? This could be made clearer.

Our reply: Yes, when the curves approach a constant value for larger values of tau, we mentioned it as the freezing phenomenon. For more clarification, we have added the following comment below Eq. (26):

We find that each of the measures initially decreases with time, then starts to increase, and finally remains approximately constant over time, a behavior known as the freezing phenomenon [91–97].”

2) Referee: In Fig. 4, in the first revival, it seems that MID does not exactly coincide with N and HSS, while it seems to for larger tau? Is there any reason for this? Check ‘non-Markovian’ in the figure caption.

Our reply: We agree with the referee that in the first revival, the MID does not exactly coincide with N and HSS, however the coincidence of HSS and entanglement is confirmed and this is enough to illustrate that HSS works well. We have clarified this behavior in the main text below Eq. (38):

“Moreover, at the first revival of the measures, the minimum point of the HSS exactly coincides with that of the negativity. After that moment we see that maximum (minimum) points of the HSS are in complete coincidence with maximum (minimum) points of the negativity as well as the MID. This perfect qualitative agreement between HSS and entanglement or quantum correlations is evidence that the HSS-based witness can precisely detect non-Markovianity in the presence of classical noises.”

We have also corrected the typo in the caption of Figure 4.

3) Referee:  In Fig. 5, it is difficult to the blue N curve for p=0.1. See also Fig. 8.

Our reply: We have made all the curves solid. Moreover, in Fig. 4, the negativity curve is amplified by 2 times for better comparison.

4) Referee: In Fig. 6, the reader is left with the impression that there is less structure in the HSS curves (red) as compared to the MID ones, for p=0.1, p=0.4 and p=0.5. Is this so, or due to the presentation? Would making all curves solid (instead of dash dot) aid in the presentation? The colors would distinguish the curves. The frame around the p=0.5 plot is boldfaced (also the inset with legend) and is not consistent with the remainder. See also Figs. 3, 5, 8.

Our reply: For common environments, we know that mutual interaction between subsystems, induced by the common environment, may lead to the preservation of correlations or even result in creation of quantum correlations between the subsystems (PHYSICAL REVIEW A 79, 042302 (2009). Therefore, revivals of quantum correlations cannot be necessarily linked to pure non-Markovianity effects and hence we do not expect complete consistency between the HSS and quantum correlations behaviors. Accordingly, Fig. 6, presenting a comparison between dynamics of quantum correlations and HSS when the subsystems are subject to a common environment, cannot be used as a sanity check example of our witness. Instead, after verification of our witness through the independent qubits, it can be viewed as an extra case for studying the HSS behavior in such systems.

We have added these comments below Eq. (42):

“For common environments, we know that mutual interaction between subsystems, induced by the common environment, may lead to the preservation of correlations or even result in creation of quantum correlations between the subsystems [106–109]. Therefore, revivals of quantum correlation cannot be necessarily linked to pure non-Markovianity effects and hence we do not expect complete consistency between the HSS and quantum correlations behaviours (see Fig. 6 demonstrating this feature of common environments causing the MID to fail in detecting the non-Markovianity). Except for these situations, we see that the maximum (minimum) points of the HSS computed for the initial pure state are very close to those of the MID calculated for the initial mixed state.”

 Moreover, we have modified all the plots according to the referee's comments. 

5) Referee

  • Check 'nonmonotonic' on line 31.
  • 6) Reference [59] is duplicated in [69].
  • See also [48] and [93].

Our reply:

  • We have corrected the typo.
  • We have eliminated duplicated references, maintaining only Refs. [59] and [48].

Reviewer 2 Report

I suggest revision according to my review here enclosed.

Author Response

Reply to Reviewer 2

We would like to thank the reviewer for the constructive comments which contributed to improve the overall presentation of the manuscript. In the following we provide our reply to the raised points.

Referee: The paper needs a serious revision to make their proposal clear. What notion of non-Markovianity are the authors investigating?

Our reply: We completely agree with the referee that we should explicitly clarify which notion of non-Markovianity is investigated in this paper. Therefore, we have added Sec. 2.1 to explain this. Moreover, at the beginning of Sec. 3, we explain that we have adopted BLP measure as our definition of non-Markovianity (page 5, lines 144-151):

In this section we check the sanity of HSS-based witness through several paradigmatic high dimensional quantum systems. The analyses are based on the fact that for systems in which the corresponding subsystems are coupled to independent environments, the oscillations of quantum correlations with time are associated with the non-Markovian evolution of the system [48,83,84], resulting in the transfer of correlations back and forth 146 among the various parts of the total system. Moreover, by comparing the results presented in references [62,85–87], we can demonstrate that the BLP measure of non-Markovianity can be used as a valid definition of non-Markovianity, when we intend to detect non-Markovianity by revivals of quantum correlations.”

Referee: In addition, they need an argument to justify the derivative appearing in the HSS definition, unless they assume that as a hypothesis or they restrict themselves to finite dimensional spaces. This is a sensible point since the authors claim that their method could be useful in high dimensional-systems (infinite-dimensional Hilbert spaces).

Our reply: We agree with the referee that our results cannot be generalized to infinite-dimensional systems in the current form. Therefore, we have highlighted that the study is restricted to high-dimensional systems with finite Hilbert spaces (see the title of Sec. 3, page 5).

Referee: Finally, a recent paper (K. Goswami et al. [1]) reports an experiment in quantum optics dynamics, leading to non-Markovianity measured through quantum relative entropy between the observed process and the associated Markovian one. I should appreciate if the authors could compare the HSS protocol and the above method. References [1] K. Goswami, C. Giarmatzi, C. Monterola, S. Shrapnel, J. Romero, and F. Costa. Experimental characterization of a non-Markovian quantum process. Physical Review A, 104(2):022432, 2021. [2] Alireza Shabani and Daniel Lidar. Vanishing quantum discord is necessary and sufficient for completely positive maps. Physical Review Letters, 102:100402, 2009. [3] Alireza Shabani and Daniel Lidar. Erratum: Vanishing quantum discord is necessary and sufficient for completely positive maps. Physical Review Letters, 116(4):049901(2), 2016. 2

Our reply: We thank the referee for pointing this out. We have added the following comment at the end of the conclusions as a promising outlook:

“Recently K. Goswami et al. [113] have reported a quantum-optics experimental setup to implement a non-Markovian process—specifically, a process with initial classical correlations between system and environment. It should be noted that in all systems investigated in this paper we have adopted the usual assumption that the system and its environment are initially uncorrelated. It would be interesting to generalize the application of the HSS-based non-Markovianity witness to scenarios in which initial correlations between the system and environment rise. This will be studied in detail in our future works.”

Reviewer 3 Report

The manuscript is devoted to the problems related to detecting non-Markovianity (nM). The Authors propose and discuss the application of the nM witness based on the Hilbert-Schmidt speed (HSS). They emphasize that the discussed method could be especially useful in research concerning high-dimensional and multipartite open quantum systems – the method does not require diagonalization of the density matrix describing the system. The Authors compare the time-evolution of HSS witness to that of the negativity and measurement-induced disturbance discussing various cases. They concentrate on the spin-S systems interacting with a thermal reservoir modeled by an infinite chain of quantum harmonic oscillators. In particular, the coupling to the independent squeezed vacuum, independent and common classical, and composite classical-quantum reservoirs are considered. The Authors prove that the proposed witness is useful not only for the low-dimensional systems (as shown in Ref. [60]) but can also be successfully applied in discussions of high-dimensional and multipartite models. Moreover, they show that HSS witness allows detecting memory effects during the time-evolution of quantum systems. Their findings are especially interesting in the context of the interpretation of nM in terms of the backflow of information from the environment to the system.

The results presented in the manuscript seem to be correct, and they are valid enough to be published. The article is well written and is accessible to a broad range of Readers. The ideas considered in the paper and obtained results are presented straightforwardly and were thoroughly discussed. I can state that the manuscript deserves publication in its present form.

Author Response

Reply to Reviewer 3

We thank the referee for the very positive overall assessment of our manuscript and for suggesting its publication in its present form.

Reviewer 4 Report

The manuscript confuses two different concepts, namely non-Markovianity and memory. 
The manuscript makes statements that have not been proven. 
When describing non-Markovian dynamics, the authors do not take into account the зщыышиду general formÑ‹ of dynamic equations, 
which are a generalization of the Lindblad equations. 
The behavior of entropy for the considered quantum systems is neither described nor discussed. 
The manuscript can be rejected. 

Author Response

Reply to Reviewer 4

Referee: "The manuscript confuses two different concepts, namely non-Markovianity and memory."

Our reply: We have clarified the definition of non-Markovianity and its relation to memory effects (associated to information backflows from environment to the system) in the new Sec. 2.1. There is also a further explanation (already included in the previous version) presented at the end of Sec. 3.3.2, page 13, lines 268-278, which should help to remove this possible ambiguity. 

Referee: "The manuscript makes statements that have not been proven. When describing non-Markovian dynamics, the authors do not take into account the зщыышиду general formÑ‹ of dynamic equations, which are a generalization of the Lindblad equations."

Our reply: Our manuscript is a thorough case study to assess the faithfulness of the HSS-based witness as a non-Markovianity witness in high-dimensional quantum systems. This extends the results presented in Ref. [62] and motivates further analyses.

Referee: The behavior of entropy for the considered quantum systems is neither described nor discussed.

Our reply: The aim of our manuscript is to validate the HSS-based non-Markovianity witness in high-dimensional open quantum systems. There is no specific interest in treating entropic quantities. As a matter of fact, the journal Entropy, and in particular the Special Issue to which the present paper has been submitted, is not limited to topics related to entropy. Our study concerns in general open quantum systems and the efficient characterization of non-Markovianity during their dynamics. This is suited for both the journal and the Special Issue.

Said so, we however would like to share some comments to the referee about entropic measures in the context of non-Markovianity. Although there are some efforts in detecting non-Markovianity through state entropy, the latter has known problems when we consider it as a general indicator: for example, it does not necessarily behave in a monotonic way even for semigroup dynamics. It should be noted that a derivation for monotonic property of the Rényi entropy under Markovian unital dynamics has been presented in [PHYSICAL REVIEW A 96, 032115 (2017)]. Moreover, non-Markovian effects on the entropy production rate in quantum thermodynamics and in the presence of a heat reservoir has been investigated [PHYSICAL REVIEW E 98, 032102 (2018)]. These aspects are not related to the scopes of our manuscript.

Round 2

Reviewer 2 Report

The authors slightly improved the mathematical background of the paper. They perform various computations of the HSS, commenting and illustrating with computer simulations in each special case. As they announce in their introduction, this paper is nothing more than a collection of examples of HSS computation. The novelty pretended is that they consider models based on higher finite dimensional spaces. Does this worth publication? One should expect the authors could go more deeply in the theoretical comparison of HSS with other witnesses of memory effects in open quantum system dynamics. It is an interesting first step, but there is room for improvement.

Reviewer 4 Report

The content of the manuscript has not changed. 
The additional text in the introduction does not interfere with the essence of the matter. 
The manuscript confuses two different concepts, namely non-Markovianity and memory (for example see "memory effects (non-Markovianity)" in page 1). 
The manuscript makes statements that have not been proven. 

Descriptions of new memory effects on quantum dynamics are practically not discussed. 

When describing non-Markovian dynamics, the authors do not take into account the general forms of dynamic equations, 
which are a generalization of the Lindblad equations. 

The behavior of entropy for the considered quantum systems is neither described nor discussed. 
The manuscript can be rejected.